# 4-Hydroxy-7-Methoxycoumarin Inhibits Inflammation in LPS-activated RAW264.7 Macrophages by Suppressing NF-κB and MAPK Activation

**DOI:** 10.3390/molecules25194424

**Published:** 2020-09-26

**Authors:** Jin Kyu Kang, Chang-Gu Hyun

**Affiliations:** Jeju Inside Agency and Cosmetic Science Center, Department of Chemistry and Cosmetics, Jeju National University, Jeju 63243, Korea; wlsrbtjsrb@naver.com

**Keywords:** 4-hydroxy-7-methoxycoumarin, macrophage, inflammation, NF-κB, MAPK

## Abstract

Coumarins are natural products with promising pharmacological activities owing to their anti-inflammatory, antioxidant, antiviral, anti-diabetic, and antimicrobial effects. Coumarins are present in many plants and microorganisms and have been widely used as complementary and alternative medicines. To date, the pharmacological efficacy of 4-hydroxy-7-methoxycoumarin (4H-7MTC) has not been reported yet. Therefore, in this study, we investigated the anti-inflammatory effects of 4H-7MTC in LPS-stimulated RAW264.7 cells as well as its mechanisms of action. Cells were treated with various concentrations of 4H-7MTC (0.3, 0.6, 0.9, and 1.2 mM) and 40 μM L-N^6^-(1-iminoethyl)-L-lysine (L-NIL) were used as controls. LPS-stimulated RAW264.7 cells showed that 4H-7MTC significantly reduced nitric oxide (NO) and prostaglandin E_2_ (PGE_2_) production without cytotoxic effects. In addition, 4H-7MTC strongly decreased the expression of inducible nitric oxide synthase (iNOS) and cyclooxygenase (COX-2). Furthermore, 4H-7MTC reduced the production of proinflammatory cytokines such as tumor necrosis factor (TNF)-α, interleukin (IL)-1β, and IL-6. We also found that 4H-7MTC strongly exerted its anti-inflammatory actions by downregulating nuclear factor kappa B (NF-κB) activation by suppressing inhibitor of nuclear factor kappa B alpha (IκBα) degradation in macrophages. Moreover, 4H-7MTC decreased phosphorylation of extracellular signal-regulated kinase (ERK1/2) and c-Jun N-terminal kinase/stress-activated protein kinase (JNK), but not that of p38 MAPK. These results suggest that 4H-7MTC may be a good candidate for the treatment or prevention of inflammatory diseases such as dermatitis, psoriasis, and arthritis. Ultimately, this is the first report describing the effective anti-inflammatory activity of 4H-7MTC.

## 1. Introduction

Coumarins (benzo-α-pyrones) are oxygen heterocycles that are naturally occurring benzopyrene derivatives which have been identified in plants, bacteria, and fungi [1]. Coumarins represent a broad family of secondary metabolites that are found naturally in over 1300 plant species. The main pathway of coumarin biosynthesis occurs through the shikimic acid pathway, which involves cinnamic acid and phenylalanine metabolism [2,3]. Natural coumarins are subdivided into several classes according to their chemical diversity and complexity, namely, simple coumarins, isocoumarins, furanocoumarins, pyranocoumarins (both angular and linear), biscoumarins, and phenylcoumarins [4].

Coumarins have several desirable features. First, they have a low molecular weight owing to their simple structures. Second, they have high solubility in most organic solvents. Third, they have high bioavailability and low toxicity. Fourth, they have various pharmacological effects such as anticoagulant, antimicrobial, anti-inflammatory, neuroprotective, antidiabetic, anticonvulsant, and antiproliferative activities [4,5,6]. These characteristics and advantages support their roles as lead compounds in drug research and development [7]. Coumarins have diverse structures owing to the different types of substitutions in their underlying structures, which can affect biological activity. Thus, the structure-system-activity-relationship of coumarin must be carefully studied [1]. 

During our ongoing screening program designed to identify modulators of skin inflammation and melanogenesis from coumarin and its derivatives, we reported that 8-methoxycoumarin increased melanogenesis via the MAPK signaling pathway [8]. In addition, we identified that auraptene, the most abundant naturally occurring geranyloxycoumarin, possesses anti-melanogenic activity through ERK-mediated MITF downregulation [9]. Furthermore, we reported that 7,8-dimethoxycoumarin stimulates melanogenesis via MAPK-mediated MITF upregulation and attenuates the expression of IL-6, IL-8, and CCL2/MCP-1 in TNF-α-treated HaCaT cells [10,11]. 

As an extension of this study, we investigated the anti-inflammatory effects of 4-hydroxy-7-methoxycoumarin (4H-7MTC, Figure 1). 4H-7MTC belongs to a class of organic compounds known as hydroxycoumarins. These are coumarins that contain one or more hydroxyl groups attached to the coumarin skeleton. 4H-7MTC can be found in plants such as coriander, artichoke, Tibetan hulless barley, and eggplant [12,13]. To the best of our knowledge, no studies have reported the pharmacological and biochemical properties and therapeutic applications of 4H-7MTC. Therefore, in this study, we investigated whether 4H-7MTC has anti-inflammatory effects; an initial step in the development of 4H-7MTC as a functional compound for use in human health applications.

## 2. Results and Discussion

Macrophages, the main cells responsible for innate immunity, are activated by the invasion of foreign pathogens such as parasites, bacteria, and viruses, or by stimulation with external signals. In particular, lipopolysaccharide (LPS), an endotoxin produced by Gram-negative bacteria, stimulates macrophages, which in turn promotes secretion of proinflammatory cytokines, including tumor necrosis factor (TNF)-α, interleukin (IL)-1β, and IL-6, and induces the expression of inflammatory response factors such as nitric oxide (NO) and prostaglandin E_2_ (PGE_2_) [14,15]. As such, regulation of the production of NO and proinflammatory cytokines in macrophages is a current research topic for the development of new anti-inflammatory agents, and there have been many attempts to derive new anti-inflammatory agents from natural compounds [16,17,18]. 

To demonstrate the anti-inflammatory activity of the three types of 4-hydroxycoumarin, including 4H-7MTC, we first assessed its ability to inhibit NO production in LPS-stimulated macrophage RAW264.7 cells (Figure 1). RAW264.7 cells were treated with various concentrations of 4-hydroxycoumarins, and cell viability was measured using the MTT assay. As shown in Figure 2, NO production increased by 3.43- to 15-fold in LPS-activated macrophages relative to untreated macrophages. Moreover, 4-hydroxycoumarins reduced LPS-induced NO production in a concentration-dependent manner. At 0.6 mM concentration of 4H-7MTC, the production of NO by LPS-treated macrophages decreased by 23.10%. At 0.5 mM concentration of 4H-6MC and 4H-7MC, the production of NO by LPS-treated macrophages decreased by 21.27% and 17.61%, respectively. These results show that the 4-hydroxy structure of coumarin influences the degree of inhibition of NO production, and the substituents on carbon 6 and 7 of the B-ring structure had little effect on the inhibition of NO production. No concentration of 4-hydroxycoumarins displayed significant cytotoxicity, indicating that the anti-inflammatory effects of 4-hydroxycoumarins were not attributable to cytotoxicity. Among them, we found that 4H-7MTC is a safe substance that does not induce cytotoxicity even at concentrations as high as 1.2 mM.

To investigate the additional functionalities of 4H-7MTC, which was confirmed to be safe at high concentrations, we aimed to evaluate its potential activity as an anticancer agent or as a preventive of gray hair. As shown in Figure 3a, 4H-7MTC upregulated melanin production in a concentration-dependent manner over a wide concentration range (25–200 μM), without any observed cytotoxicity. Additionally, 4H-7MTC showed no cytotoxicity up to 1.2 mM in normal macrophages, whereas it exhibited a cytotoxic effect on B16F10 melanoma cells at a low concentration of 0.3 mM (Figure 3b). This suggests that 4H-7MTC could be a potential anticancer agent.

To further elucidate the anti-inflammatory mechanisms of 4H-7MTC, we measured the levels of PGE_2_, IL-6, IL-1β, and TNF-α in culture supernatants using ELISA. Treatment of RAW264.7 cells with LPS alone resulted in a significant increase in cytokine production compared to that in the drug groups (Figure 4). However, NO, PGE_2_, IL-6, IL-1β, and TNF-α levels in the supernatants of LPS-stimulated cells pretreated with 0.3, 0.6, 0.9, and 1.2 mM 4H-7MTC were significantly reduced compared to those in the LPS group in a concentration-dependent manner (Figure 4).

To further elucidate the mechanisms by which 4H-7MTC inhibited NO and PGE_2_ production in LPS-activated macrophages, we analyzed the effects of 4H-7MTC on LPS-induced iNOS and COX-2 gene expression in macrophages. Under normal conditions, RAW264.7 cells expressed non-detectable levels of COX-2 expression, but iNOS and COX-2 protein levels markedly increased after 18 h of LPS stimulation (Figure 5). With the addition of 4H-7MTC (0.3, 0.6, 0.9, and 1.2 mM), concentration-dependent inhibition of iNOS and COX-2 expression was observed, indicating that 4H-7MTC modulates iNOS and COX-2 expression. 

A previous study revealed that NF-κB activation in response to pro-inflammatory stimuli involves the rapid phosphorylation of IκBs by the IKK signalosome complex. Free NF-κB produced by this process translocates to the nucleus where it binds to κB-binding sites in the promoter regions of target genes. It then induces the transcription of pro-inflammatory mediators such as iNOS and COX-2. Several studies have shown that anti-inflammatory agents inhibit NF-κB activation by preventing IκB degradation [19,20,21]. Thus, we attempted to determine whether 4H-7MTC inhibits IκB phosphorylation and degradation. Accordingly, RAW264.7 cells were pretreated for 1 h with 4H-7MTC, and IκB-α protein levels were determined after 20 min of LPS exposure (1 μg/mL). As shown in Figure 6, 4H-7MTC significantly suppressed LPS-induced phosphorylation and degradation of IκB-α. These results show that 4H-7MTC inhibits LPS-induced NF-κB activation by preventing the degradation of IκB-α phosphorylation.

MAPK plays a critical role in regulating cell growth and differentiation and controls cellular responses to cytokines and stress. In addition, three MAP kinases (JNK, p38 MAPK, and ERK 1/2) have been reported to be adjustable in LPS-induced pro-inflammatory cytokine production [22,23,24,25].

To investigate the molecular mechanism of MAPK signaling by 4H-7MTC in LPS-stimulated RAW264.7 cells, we studied the inhibition of phosphorylation of ERK1/2, p-38, and JNK. RAW264.7 cells were pretreated with 4H-7MTC at the indicated concentrations for 1 h and then stimulated with 1 µg/mL LPS for 1 h. The total cell lysates were then probed with phosphospecific antibodies for ERK1/2 and JNK. Phosphorylation of ERK1/2 and JNK increased in cells treated with LPS alone. Pretreatment with 4H-7MTC inhibited the LPS-induced phosphorylation of JNK and ERK 1/2 in a concentration-dependent manner, but not that of p38 MAPK. The amount of non-phosphorylated MAPKs was not affected by either LPS or 4H-7MTC treatment (Figure 7). 

These results suggest that suppression of MAPK phosphorylation may be involved in the inhibitory effect of 4H-7MTC on LPS-stimulated inflammatory response factors and inflammatory cytokines via NF-κB signaling in RAW264.7 cells.

## 3. Materials and Methods

### 3.1. Chemicals and Reagents

4-Hydroxy-7-methoxycoumarin (4H-7MTC), 4-Hydroxy-6-methylcoumarin (4H-6MC), and 4-Hydroxy-7-methylcoumarin (4H-7MC) were obtained from Tokyo Chemical Industry (Tokyo, Kita-ku, Japan). Lipopolysaccharide (LPS) from *Escherichia coli*, 3-(4,5-dimethylthiazol-2-yl)-2,5-diphenyltetrazolium bromide (MTT), α-melanocyte-stimulating hormone (α-MSH), dimethyl sulfoxide (DMSO), Griess reagent, sodium nitrite, and protease inhibitor cocktail were obtained from Sigma-Aldrich (St Louis, MO, USA). Dulbecco’s Modified Eagle Medium (DMEM), fetal bovine serum, and penicillin/streptomycin were obtained from Thermo Fisher Scientific (Waltham, MA, USA). Radioimmunoprecipitation assay buffer, phosphate-buffered saline (PBS), enhanced chemiluminescence (ECL) kit, and tris-buffered saline (TBS) were obtained from Biosesang (Seongnam, Gyeonggi-do, Korea). *N*-[2-(Cyclohexyloxy)-4-nitrophenyl] methanesulfonamide (NS-398), and L-N^6^-(1-iminoethyl) lysine dihydrochloride (L-NIL) were obtained from Cayman Chemical Company (Ann Arbor, MI, USA). Prostaglandin E_2_ (PGE_2_) ELISA kit, interleukin-1β (IL-1β) kit, IL-6 ELISA kit, and tumor necrosis factor (TNF-α) ELISA kits were obtained from R&D System Inc. (St. Louis, MO, USA). The following antibodies were used in this study: β-actin, anti-iNOS, anti-inhibitor of NF-κB (IκBα), Akt, p-Akt, p38, p-p38, JNK, p-JNK, ERK, and p-ERK were obtained from Cell Signaling Technology (Beverly, MA, USA). Anti-COX-2 was obtained from BD Biosciences (San Diego, CA, USA). All reagents used were of analytical grade.

### 3.2. Cell Culture

RAW264.7 mouse macrophages and B16F10 melanoma cells were obtained from the Korean Cell Line Bank (Seoul, Korea). RAW264.7 cells were subcultured at intervals of 2–3 days. The B16F10 melanoma cells were subcultured at 4-day intervals using DMEM with 10% FBS, 100 U/mL penicillin, and 100 μg/mL streptomycin at 37 °C in a humidified 5% CO_2_ atmosphere.

### 3.3. Cell Viability

Cytotoxicity was determined using the MTT assay. RAW264.7 cells were cultured at a density of 1.5 × 10^5^ cells/well in 24-well plates for 24 h. Cells were treated with various concentrations of 4H-7MTC (0.3, 0.6, 0.9, and 1.2 mM). RAW264.7 cells were incubated for 24 h and MTT solution (0.2 mg/mL) was added to the medium and incubated for 4 h. Next, the medium was removed and formazan crystals in each well were dissolved in DMSO for 20 min. Optical density (OD) was measured at 570 nm, and the percentage of cells showing cell viability relative to the control was determined.

### 3.4. NO Production

NO production in the cell culture was assayed by measuring the accumulated nitrite using Griess reagent. RAW264.7 cells were plated at a density of 1.5 × 10^5^ cells/well in 24-well plates. Cells were pretreated with various concentrations of 4H-7MTC (0.3, 0.6, 0.9, 1.2 mM) for 1 h and treated with LPS (1 μg/mL) for 24 h. Then, the treated cell culture solution was mixed with the Griess reagent in a 1:1 ratio, reacted for 15 min, and the absorbance measured at 540 nm using a spectrophotometer. NO production in the sample was quantified from a standard curve constructed using sodium nitrite.

### 3.5. Measurement of Cytokines

RAW264.7 mouse cells were plated at a density of 1.5 × 10^5^ cells/well in 24-well plates. Cells were pretreated with various concentrations of 4H-7MTC (0.3, 0.6, 0.9, and 1.2 mM) for 1 h and treated with LPS (1 μg/mL) for 24 h. Supernatants were harvested, and PGE_2_, IL-1β, IL-6, and TNF-α levels were measured using ELISA kits according to the manufacturer’s protocols.

### 3.6. Measurement of Melanin Content

B16F10 melanoma cells were plated in 60 mm cell culture dishes (6.0 × 10^4^ cells/dish), incubated for 24 h, and then treated with 4H-7MTC (25, 50, 100, 150, and 200 μM) for 72 h in the presence of α-MSH (100 nM). After incubation, the cells were washed with 1 × PBS and the pellets were solubilized in 1 N NaOH containing 10% DMSO at 70 °C for 1 h. Absorbance was measured at 405 nm with a spectrophotometer. The protein concentration was determined using a BCA protein analysis kit.

### 3.7. Western Blot Analysis

RAW264.7 mouse cells were plated at a density of 6.0 × 10^5^ cells/dish in 60-mm cell culture dishes for 24 h. Cells were pretreated with various concentrations of 4H-7MTC (0.3, 0.6, 0.9, 1.2 mM) for 1 h and treated with LPS (1 μg/mL) for the indicated times. After incubation, cells were washed with 1 × PBS and lysed on ice with RIPA lysis buffer (150 mM NaCl, 50 mM Tris-HCl [pH 7.5], 2 mM EDTA, 1% Triton X-100, 0.1% SDS, and 1% protease inhibitor cocktail) for 30 min. The harvested cell lysates were centrifuged at −8 °C and 15,000 rpm for 20 min. A standard assay curve of bovine serum albumin (BSA) was prepared using the BCA Protein Assay Kit, and the protein contents of the extracted cell lysates were quantitatively determined. The protein concentration was determined using a BCA protein analysis kit. Whole-cell lysates (30 μg) were separated by SDS-polyacrylamide gel electrophoresis on a 10% gel (SDS-PAGE) and electroblotted onto polyvinylidene fluoride (PVDF) membranes. The membranes were then blocked with 5% skim milk and incubated for 2 h. The membrane was washed 6 times with TBS buffer containing 0.1% Tween 20 (TTBS) and then incubated with specific primary antibodies (1:2500) at 4 °C for 6 h. The membrane was washed 6 times with TTBS buffer and incubated with a peroxidase-conjugated secondary antibody (1:2000) at room temperature for 2 h. The membrane was then washed six times with TTBS buffer and the protein was detected using an ECL kit.

### 3.8. Statistical Analysis

All results are expressed as mean ± standard deviation (SD). Each value represents the mean of three independent experiments. Statistical analysis was performed using a one-way analysis of variance (ANOVA) followed by Tukey’s post-hoc test, and survival rates between multiple groups were analyzed using the log-rank test. The significant difference was set at * *p* < 0.05, ** *p* < 0.01, and *** *p* < 0.001.

## 4. Conclusions

This study is, to the best of our knowledge, the first to elucidate the anti-inflammatory properties of 4H-7MTC, which was mediated through the suppression of NO, PGE_2_, IL-6, IL-1β, and TNF-α production in LPS-stimulated RAW264.7 cells via the NF-κB and MAPK signaling pathways. Our findings indicate that 4H-7MTC may be a promising agent for the clinical prevention and treatment of inflammation-associated diseases in the future. Additionally, 4H-7MTC has also been shown to enhance melanin production and has a potential application as an anticancer agent.

## Figures and Tables

**Figure 1 molecules-25-04424-f001:**
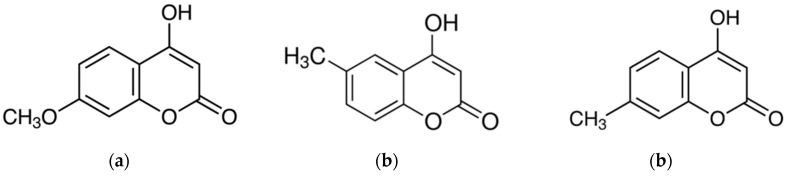
Structures of 4-hydroxycoumarins: 4-hydroxy-7-methoxycoumarin (**a**), 4-hydroxy-6-methylcoumarin (**b**), and 4-hydroxy-7-methylcoumarin (**c**).

**Figure 2 molecules-25-04424-f002:**
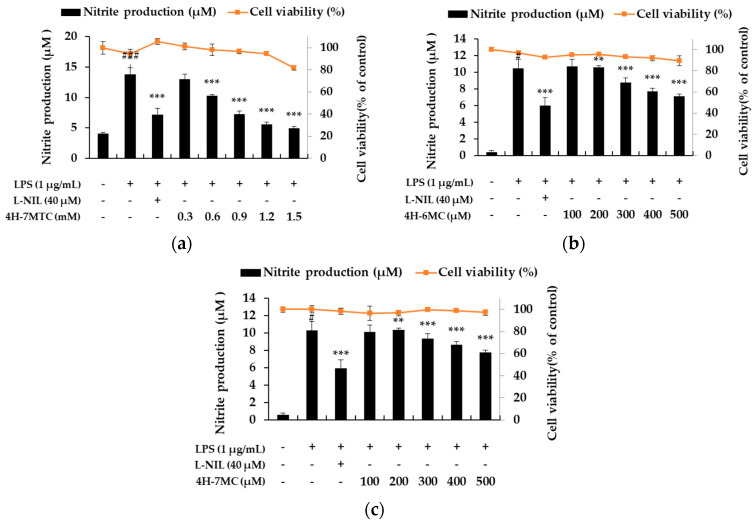
Effect of 4H-7MTC (**a**), 4H-6MC (**b**), and 4H-7MC (**c**), on nitric oxide production in LPS-stimulated RAW264.7 cells. The cells were plated in 24-well plates (1.5 × 10^5^ cells/well), incubated for 24 h, and then pretreated with 4H-7MTC (0.3, 0.6, 0.9, 1.2, and 1.5 mM), 4H-6MC (100, 200, 300, 400, and 500 μM), and 4H-7MC (100, 200, 300, 400, and 500 μM) for 1 h, followed by LPS stimulation for 24 h. Cytotoxicity of 4H-7MTC, 4H-6MC, and 4H-7MC were evaluated using MTT assay. Nitric oxide production was determined by the Griess reagent method. L-N6-(1-Iminoethyl) lysine dihydrochloride (L-NIL) was used as a positive control. The data are presented as mean ± SD. Statistical significance was assessed by one-way analysis of variance (ANOVA), followed by Tukey’s post-hoc test and represented as follows: ^#^
*p* < 0.05, ^###^
*p* < 0.005, ** *p* < 0.01, *** *p* < 0.001 vs. LPS alone.

**Figure 3 molecules-25-04424-f003:**
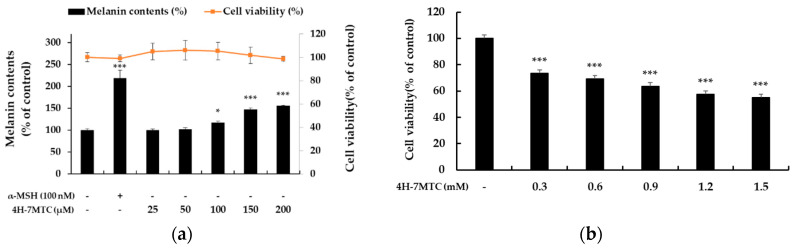
Effect of 4H-7MTC on the production of melanin (**a**) in α-MSH-stimulated B16F10 cells and Cytotoxicity of 4H-7MTC in B16F10 cells (**b**). Cells were plated in 60 mm cell culture dish (6.0 × 10^4^ cells/dish), incubated for 24 h, and then treated with 4H-7MTC (25, 50, 100, 150 and 200 μM) for 72 h in the presence of α-MSH (100 nM). α-MSH was used as the negative control. Cytotoxicity of 4H-7MTC was evaluated using MTT assay. Cells were plated in 24-well plates (1.5 × 10^4^ cells/well) for 24 h, and then treated with 4H-7MTC (0.3, 0.6, 0.9, 1.2, and 1.5 mM) for 72 h. The data are presented as mean ± SD. Statistical significance was assessed by one-way analysis of variance (ANOVA) followed by Tukey’s post-hoc test and represented as follows: * *p* < 0.05, *** *p* < 0.001 vs. LPS alone.

**Figure 4 molecules-25-04424-f004:**
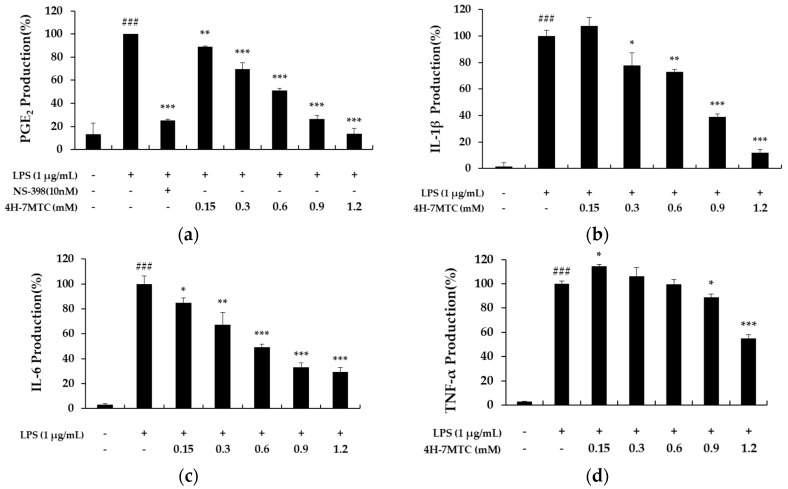
The effect of 4-hydroxy-7-methoxycoumarin (4H-7MTC) on the LPS-induced production of proinflammatory cytokines in RAW264.7 cells. Cells were pretreated with 4H-7MTC (0.15, 0.3, 0.6, 0.9, and 1.2 mM) for 1 h and then stimulated for 20 h with LPS. The production of PGE_2_ (**a**), IL-1β (**b**), IL-6 (**c**), and TNF-α (**d**) were determined using ELISA. The data are presented as the mean ± SD. Statistical significance was assessed by one-way analysis of variance (ANOVA) followed by Tukey’s post-hoc test and represented as follows: Values are representative of three independent experiments. ^###^ p < 0.005 vs. control cells. * *p* < 0.05, ** *p* < 0.01, *** *p* < 0.001 vs. LPS alone.

**Figure 5 molecules-25-04424-f005:**
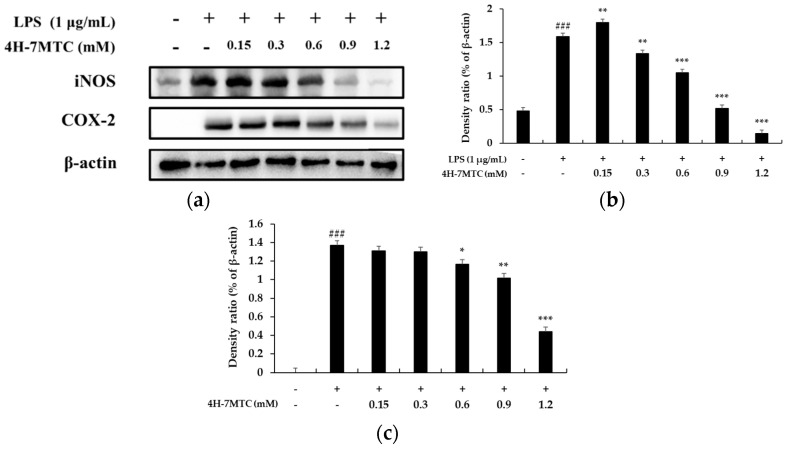
Effect of 4-hydroxy-7-methoxycoumarin (4H-7MTC) on the level of iNOS in LPS-induced RAW264.7 cells. Lysates were prepared from cells pretreated with 4H-7MTC (0.15, 0.3, 0.6, 0.9, and 1.2 mM) for 1 h and treated with LPS (1 μg/mL) for 18 h. β -actin was used as a loading control. Total cellular proteins were separated using SDS-PAGE, transferred to PVDF membranes, and detected using specific antibodies against iNOS and β-actin (**a**). Results are presented as representative of three independent experiments and summarized in the bar graphs (**b**,**c**). ^###^
*p* < 0.005 vs. control cells. * *p* < 0.05, ** *p* < 0.01, *** *p* < 0.005 vs. LPS-induced cells.

**Figure 6 molecules-25-04424-f006:**
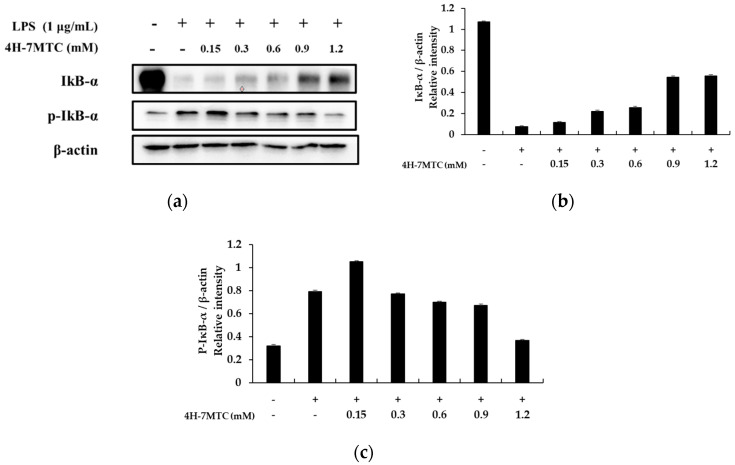
Effect of 4-hydroxy-7-methoxycoumarin (4H-7MTC) on the level of phospho-IκB-α and IκB-α in LPS-induced RAW264.7 cells. Lysates were prepared from cells pretreated with 4H-7MTC (0.15, 0.3, 0.6, 0.9, and 1.2 mM) for 1 h and then treated with LPS (1 μg/mL) for 20 min. Western blotting was performed to detect the expression of IκBα and p-IκBα. β-actin was used as a loading control (**a**). Quantification of immunoreactive protein bands is shown via bar graphs (**b**,**c**).

**Figure 7 molecules-25-04424-f007:**
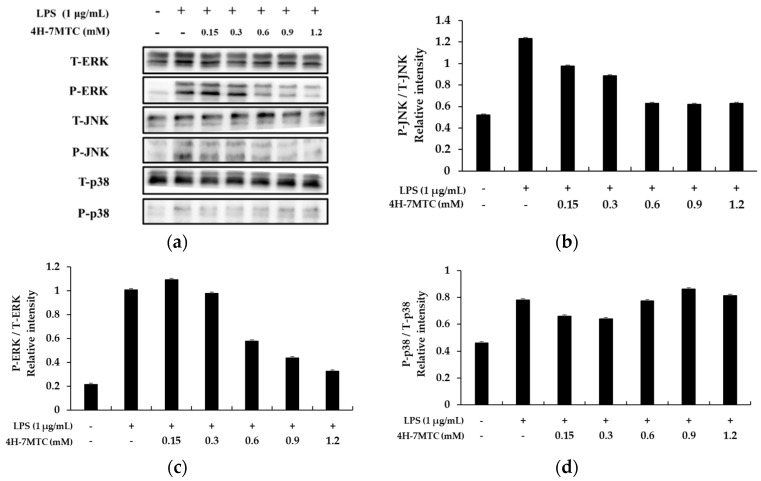
Effect of 4-hydroxy-7-methoxycoumarin (4H-7MTC) on LPS-induced MAPK in RAW264.7 cells. Lysates were prepared from cells pretreated with 4H-7MTC (0.15, 0.3, 0.6, 0.9, and 1.2 mM) for 1 h and treated with LPS (1 μg/mL) for 15 min. Western blotting was performed to detect the expression of phospho-ERK, T-ERK, phospho-JNK, T-JKN, phospho-p38, and T-p38 (**a**). β-actin was used as a loading control. Quantification of immunoreactive protein bands is shown via bar graphs (**b**–**d**).

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
