# Peer review of "4-Hydroxy-7-Methoxycoumarin Inhibits Inflammation in LPS-activated RAW264.7 Macrophages by Suppressing NF-κB and MAPK Activation"

_molecules, 2020, doi:10.3390/molecules25194424_

Round 1

Reviewer 1 Report

The manuscript reports on the anti-inflammatory activity of one 4-hydroxy-7-methoxycoumarin. It is a purely pharmacological study. No chemical investigation has been reported.
The biological results are interesting, but the manuscript could be a better fit at a Journal which has a more pharmacological aim in its scope.

The manuscript “4-Hydroxy-7-methoxycoumarin Inhibits 3 Inflammation in LPS-activated RAW264.7 4 Macrophages by Suppressing NF-κB and MAPK 5 Activation” reports on the bioactivity evaluation of only one commercially available coumarin (4-hydroxy-7-methoxycoumarin). I think that the manuscript could be better fit in a periodic with pharmacological scope. The results of the anti-inflammatory assays are good, but no chemical data have been provided, which certainly would be interesting for the Molecules´ readers. Some improvements could be made: abstract and introduction should highlight the natural occurrence of the evaluated coumarin (if there is) or how it can be synthesized. These reflections could be interesting for the chemistry community. Some terms should be avoided, such as “green plants” (line 34). The authors should provide a brief discussion about the importance of screening new anti-inflammatory leads and the correlation between the inflammatory process with other diseases. Despite mentioning that the 4-hydroxy-7-methoxycoumarin has never been pharmacologically evaluated yet, the authors assayed it only for its anti-inflammatory properties. Why the authors have selected the 4-hydroxy-7-methoxycoumarin for anti-inflammatory assays?

Author Response

Thank you for your useful comments and suggestions on the language and structure of our manuscript. We have modified the manuscript accordingly, and detailed corrections are listed below point by point. Our authors received English proofreading through native speakers before resubmitting paper.

(1) Some improvements could be made: abstract and introduction should highlight the natural occurrence of the evaluated coumarin (if there is) or how it can be synthesized.

→ As pointed out by the reviewer, we have inserted basic information about the 4H-7MTC in the text; “4H-7MTC belongs to the class of organic compounds known as hydroxycoumarins. These are coumarins that contain one or more hydroxyl groups attached to the coumarin skeleton. 4H-7MTC can be found in coriander, artichoke, Tibetan hulless barley, and eggplant.”

(2) Some terms should be avoided, such as “green plants” (line 34).

→ As the reviewer points out, we have removed the term "green plants“

(3) the authors assayed it only for its anti-inflammatory properties. Why the authors have selected the 4-hydroxy-7-methoxycoumarin for anti-inflammatory assays?

→ As shown in the revised Fig. 1, we compared the NO inhibitory activity for three types of 4-hydroxycoumarin, including 4-hydroxy-7-methoxycoumarin, but there was no significant difference in NO inhibitory activity. However, in the case of 4-hydroxy-7-methoxycoumarin, we confirmed that it increases melanin production in addition to its anti-inflammatory effect, and for this reason, 4-hydroxy-7-methoxycoumarin with dual function was selected as the subject of the primary study.

Reviewer 2 Report

The article entitled “4-Hydroxy-7-methoxycoumarin Inhibits Inflammation in LPS-activated RAW264.7 Macrophages by Suppressing NF-kB and MAPK Activation” describe utilization of given coumarine’s derivative as anti-inflammation agent. The design of research is well organised and appropriate. On the other hand there is a handful of cumarine derivatives which were already tested as anti-inflammatory agents and thereof it is not surprising that this compound has some effect. Moreover, biological activity of presented compound as anti-inflammatory agent is bad when compared with plenty of known simple coumarin derivatives.

But generally, presented paper is comprehensive study of given compound as anti-inflammatory agent and it enlarge the knowledge about coumarines.

I suggest to improve figures description. For figure 3 is clearly described what is on picture a, d, c, etc. This is missing on the other figures where multiple graphs are presented.

I aslo suggest to reformulated the conclusion where should be clearly mentioned that 4H-7MC it self is not the good canditate as anti-inflammatory agent but it can further develop.

Author Response

Thank you for your useful comments and suggestions on the language and structure of our manuscript. We have modified the manuscript accordingly, and detailed corrections are listed below point by point. Our authors received English proofreading through native speakers before resubmitting paper.

(1) On the other hand there is a handful of cumarine derivatives which were already tested as anti-inflammatory agents and thereof it is not surprising that this compound has some effect

→ We totally agree with the reviewer's point. The anti-inflammatory effects of 4-H-6MC may not be surprising in itself. However, 4-H-7MTC can be used to develop anti-gray hair products by increasing melanin production. In addition, I would like to emphasize that it is an important material that is relatively safe for macrophage cells while inhibiting the growth of malanoma cells.

(2) I suggest to improve figures description. For figure 3 is clearly described what is on picture a, d, c, etc. This is missing on the other figures where multiple graphs are presented.

→ As the reviewer pointed out, we improved the figures description

(3) I aslo suggest to reformulated the conclusion where should be clearly mentioned that 4H-7MC it self is not the good canditate as anti-inflammatory agent but it can further develop.

→ As pointed out by the reviewer, we mentioned in the "Conclusion" section that 4H-7MC is a material with continuous research value.

Reviewer 3 Report

Review molecules-916895

Abstract:

Clearly sets out the reasons and central findings of the study. Identifies the unique aspect of the study.

Introduction:

Clear and steps through the main hypothesis and unique aspect of the study.

line 49, page 2 “Coumarins, which have simple structures, have pharmacological and biochemical properties as well as therapeutic applications, which depend on the pattern of substitution at their carbon and hydroxy  group positions. “ this sentence is redundant.

Methods:

Extensive and generally appropriate. Model cell line appropriate.

Statistics:

T-tests should not be used for the statistics, the appropriate test is one way ANOVA with Dunnet’s test vs control or Sidak’s test for multiple comparisons. The statistical tests need to be mentioned in the figure legends.

Results and discussion: Understandable and well presented. Data presented and explained logically. Inferences logical and not overstated.

Tables, Figures and figure legends:

Generally appropriate. In the figures, the statistical test needs to be given. Eg. One Way ANOVA *, P <0.05 etc. Figures 5 on have no statistical information in the figure legends nor significance stars on the graphs despite the results section saying that significant effects were found.

Conclusions: Appropriate.

Minor issues: replace “dose-dependent” with “concentration-dependent” throughout since defined concentrations were used, not doses given to an animal.

Author Response

Thank you for your useful comments and suggestions on the language and structure of our manuscript. We have modified the manuscript accordingly, and detailed corrections are listed below point by point. Our authors received English proofreading through native speakers before resubmitting paper.

(1) line 49, page 2 “Coumarins, which have simple structures, have pharmacological and biochemical properties as well as therapeutic applications, which depend on the pattern of substitution at their carbon and hydroxy group positions. “ this sentence is redundant.

→ As pointed out by the reviewer, we have deleted the above sentence.

(2) Tables, Figures and figure legends: Generally appropriate. In the figures, the statistical test needs to be given. Eg. One Way ANOVA *, P <0.05 etc. Figures 5 on have no statistical information in the figure legends nor significance stars on the graphs despite the results section saying that significant effects were found.

→ As the reviewer pointed out, we corrected the statistical problem of the picture.

(3) Minor issues: replace “dose-dependent” with “concentration-dependent” throughout since defined concentrations were used, not doses given to an animal.

→ As the reviewer points out, we have replaced “dose-dependent” with “concentration-dependent”

Round 2

Reviewer 1 Report

Authors have provided all recommended changes, which contributed to the improvement of the manuscript

The manuscript may be accepted

Author Response

Our authors thank the reviewers for their generous decision. It is expected to be of great help to the future of the first author who is an undergraduate student.

Reviewer 3 Report

The Authors have made extensive revisions to the manuscript which have improved it enormously. The statistics have been extensively revised and are now appropriate.  However, one again Figures 5 on have no statistical information in the figure legends nor significance stars on the graphs despite the results section saying that significant effects were found (indeed panels b), c) and in the case of fig.7 d) are not even mentioned). The log-rank test is mentioned in the statistics section but does not appear anywhere else in the manuscript.

The authors state (page 2, line 58) “As an extension of this study, we investigated the effects of 4-hydroxy-7-methoxycoumarin (4H-58 7MTC) on human health.” The authors have only studied cells in culture, this has implications for human health, but is not a study on human health. This sentence needs rewording.

Author Response

We are always pointed out about statistical information. We confirmed the significance of each through repeated experiments, but we are struggling with immature statistical processing. If the reviewers point out in detail, we will check the data again.
Meanwhile, as pointed out by the reviewers, we have modified the term "human health" to "anti-inflammatory effects".